# Global Governance of Front-of-Pack Nutrition Labelling: A Qualitative Analysis

**DOI:** 10.3390/nu11020268

**Published:** 2019-01-25

**Authors:** Anne Marie Thow, Alexandra Jones, Carmen Huckel Schneider, Ronald Labonté

**Affiliations:** 1Menzies Centre for Health Policy, Sydney School of Public Health, University of Sydney, Sydney, NSW 2006, Australia; carmen.huckelschneider@sydney.edu.au; 2The George Institute for Global Health, Sydney, NSW 2042, Australia; ajones@georgeinstitute.org.au; 3School of Epidemiology and Public Health, University of Ottawa, Ottawa, ON K1G 5Z3, Canada; rlabonte@uottawa.ca

**Keywords:** nutrition labelling, codex, policy, noncommunicable disease prevention

## Abstract

The Codex Alimentarius has approved ongoing work for international guidance on front-of-pack (FoP) nutrition labelling, which is a core intervention for prevention of diet-related noncommunicable disease. This guidance will have implications for national policy decision-making regarding this important public health issue. However, FoP nutrition labelling is also a trade and commerce policy issue. In this study, we analyze the global governance of FoP nutrition labelling and current policy processes, to inform public health policy and advocacy. We present findings from a qualitative governance and institutional analysis, based on key informant interviews with 28 global actors. The study found that Codex guidance was perceived as likely to have a high impact on FoP nutrition labelling globally. However, a small and highly interconnected “regime complex” of international institutions surrounds FoP nutrition labelling at the global level, and influence on Codex discussions is being exerted differentially by actors at the national and global level, particularly by government and industry actors. There are thus risks associated with conflicts of interests in the development of global guidance on FoP nutrition labelling. There are also opportunities for more strategic and coordinated public health engagement.

## 1. Introduction

Front-of-pack (FoP) nutrition labelling is a core component of the emerging “essential” package of policy recommendations to address the growing global burden of diet-related noncommunicable diseases (NCDs) [1]. Diet-related ill-health contributes substantially to the global burden of disease, particularly excess consumption of harmful fat, salt and sugar, and inadequate intake of fruits, vegetables, and other minimally processed foods [2]. Existing nutrition information, typically appearing on the back of pack, have proved of limited value in helping consumers to understand the relative healthfulness of packaged food. An increasing body of research indicates that interpretive FoP nutrition labels, such as “traffic-light” labels adopted by some retailers in the United Kingdom, the “Health Star Rating” system implemented in Australia, or the “warning labels” implemented in Chile, can be effective in stimulating healthier choices at the point of purchase [3,4]. There is also some evidence that such labels can stimulate companies to reformulate products towards healthier nutrient compositions [5,6].

As interest in—and implementation of—FoP nutrition labelling for NCD prevention has increased at a country-level, challenges have become apparent. Diverse approaches have been taken to FoP nutrition labelling, which partly reflect an emerging evidence base and partly contextual differences (cultural, political, and population-based). In particular, the schemes that have been implemented vary in terms of (1) designs and content, with some signposting “high” content of nutrients associated with NCD risk and others evaluating and summarizing the nutritional quality of products overall; (2) the type of judgement made (positive and/or negative judgements, such as endorsement logos or warning labels); (3) implementation mode (voluntary or mandatory) [7]. This lack of harmonization has resulted in the need for food industry actors to cater to different labelling requirements in different markets, even within the same trading region. Some mandatory FoP labelling initiatives have been subject to specific trade concerns, raised in the Technical Barriers to Trade Committee in the World Trade Organization [8].

Trade policy usually relies on standards to guide definitions of what constitutes the “necessary” and/or “least trade restrictive” requirements on traded goods. In the case of food, the Codex Alimentarius Commission constitutes the internationally recognized standards-setting body. Codex is an intergovernmental body and the principal organ of the joint World Health Organization (WHO)/Food and Agriculture Organization (FAO) food standards program. Codex was created in 1963 to develop standards to “protect consumer health and promote fair practices in food trade” [9]. The function and purpose of Codex “is to guide and promote the elaboration and establishment of definitions and requirements for food, to assist in their harmonization and, in doing so, facilitate international trade”. The adopted food standards and other texts together form the Codex Alimentarius.

Codex membership is open to all countries that are members of either of the two parent organizations, and currently stands at 188 member states and 1 member organization (the European Union (EU)) [10]. Each Member of the Commission shall have at least one representative, entitled to one vote in decision-making [10]. In addition, each member “may be accompanied by one or more alternatives and advisers” (here, ‘alternatives’ refer to additional people who can act as the representative). Codex meetings are also attended by eligible International Non-Governmental Organizations who have been approved “Observer Status”. There are currently 219 Codex observers: 56 intergovernmental organizations, 147 non-government organizations (including private sector and civil society groups) and 16 UN organizations. Observers receive privileges including the right to send representatives and advisers to attend meetings (without the right to vote), to receive all working documents and discussion papers, to circulate their views in writing to the Commission, and to participate in discussions when invited [10]. Observer collaboration is intended to provide Codex with expert information, advice, and assistance, and to enable representatives of professional and technical authorities to express their views and “play an appropriate role” in ensuring the harmonizing of intersectoral interests. In practice these non-government organizations are disproportionately industry bodies; around 75% represent industry interests [11].

Codex is explicitly referenced by the World Trade Organization (WTO) Agreement on Sanitary and Phytosanitary Measures, and meets the criteria for a standards-setting body in the WTO Agreement on Technical Barriers to Trade (TBT). Codex became more influential and more politicized after recognition by WTO agreements in 1994, as a result of increased awareness by member states that decisions in Codex may become effectively binding under the WTO agreements [12,13]. While Codex articulates its standards as a useful “floor”—the basis for potentially more stringent national standards—the WTO language in effect made Codex standards more like a “ceiling”. Effectively, this means that countries that introduce a labelling standard that is stricter that that outlined in the Codex Alimentarius can be required to justify their policies in trade forums. Some have argued that this equates to “voluntarism under duress” [14,15]. Effectively, recognition by the Agreements of the WTO significantly increased the legitimacy of Codex as the global standards-setting body for food [16,17], and thus its influence over national food standards development.

Codex has traditionally focused on issues of food safety and related issues of acute exposure to harmful substances. Codex was first provided a mandate to work on NCDs in the 2004 WHO Global Strategy on Diet, Physical Activity, and Health. Following this, Codex developed Nutrient Reference Values (NRVs) for NCDs for saturated fat and sodium and included these in the Guidelines on Nutrition Labelling in 2011 [18]. NRVs are recommendations for nutritional intake based on currently available scientific knowledge and usually relate to adequate intakes to prevent nutrient deficiency diseases or disorders. “NRVs-NCD” differ considerably since they are based on levels of nutrients associated with the reduction in the risk of diet-related noncommunicable diseases. In 2015, discussions began for the inclusion of an additional NRV-NCD for EPA + DHA (long chain omega-3 fatty acids) [19].

Existing nutrition labelling guidelines at Codex do not explicitly provide guidance on interpretive labelling. Codex guidelines for ingredients lists, nutrient content presentation and health/nutrition claims is clear and detailed [19]. In contrast, additional means of presentation of nutrition information “based on the needs of consumers” is at the discretion of “competent authorities”. Similarly, “supplementary nutrition information” is “optional” and must only be given in addition to a nutrient declaration. An exception, however, is made for target populations with high illiteracy rates—including nutrition illiteracy—but no detail is provided regarding the “food group symbols or other pictorial or color presentations” that may be used.

Discussions are currently underway at Codex regarding the potential development of guidance on FoP nutrition labelling. An electronic Working Group (eWG) was established by the Codex Committee on Food Labelling to consider FoP nutrition labelling in 2016, chaired by Costa Rica and New Zealand (Figure 1 shows the structure of Codex and relationship to other key institutions). The role of the eWG is to make recommendations for consideration by the Committee on Food Labelling, which in turn provides recommendations to the Codex Alimentarius Commission. The establishment of the eWG was also preceded by a recommendation for Codex to consider developing guidance on interpretive labelling made by the International Association of Consumer Food Organizations in May 2016, with the purpose of protecting and promoting public health [20]. The first discussion paper of the eWG included a stock take of current FoP nutrition labelling schemes, which was submitted in 2017 to the Codex Committee on Food Labelling with a recommendation that further work be undertaken. In July 2018, the Codex Alimentarius Commission formally approved work towards a guideline on FoP nutrition labelling. A second working paper sought to establish a definition for what is considered FoP for the purposes of this work.

Such a shift from national governance of interpretive nutrition labelling to formal global governance raises questions about the implications for public health. Global policy outcomes reflect the exercise of power within rules-based institutions [21]. Analysis of power and other political-economy factors has been identified as critical in strengthening health policy [22]. In relation to food security, Margulis has documented the importance of global institutional structures and mandates in shaping global action on food security [23]. In this study, we therefore seek to explicitly consider the roles of institutional structures and the exercise of power by different actors in shaping the ongoing Codex discussions on FoP nutrition labelling. Given the high likelihood that Codex guidance will be influential for national policy-making in member states—which constitute nearly all United Nations members—this study will provide valuable insights into the processes underlying global decision-making regarding FoP nutrition labelling, and thus identify opportunities for increasing attention to public health considerations and maintaining national autonomy for countries to implement health policies they consider appropriate for their own populations.

## 2. Materials and Methods

### 2.1. Study Design and Frameworks

We conducted a qualitative policy and governance analysis, based on interviews with 28 stakeholders regarding ongoing discussions at Codex, including relevant subcommittees, regarding FoP nutrition labelling. The aim of our study was to increase understanding of how decisions about interpretive nutrition labelling are made and influenced at the global level, and what the potential outcomes might be for public health nutrition. The primary research question for this study was thus: *How are decisions about FoP nutrition labelling made and influenced at the global level?* We drew on theories of governance and policy-making to inform the study design, development of the semi-structured interview guide, and data analysis. In particular, new institutional theory [24] and frame-critical analysis [25] enable action-oriented policy analysis by identifying the importance of: framing of the policy issue and perceptions of “ideal” policy content; roles, structures and relationships of actors and institutions (including power dynamics); actor interests; and processes for decision-making. Data were collected from purposefully sampled knowledgeable policy actors through semi-structured interviews, coded thematically, and analyzed with reference to the theories underpinning the study, and to different forms of power in governance, as synthesized by Haugaard [26].

### 2.2. Data Collection

We conducted semi-structured interviews with 28 stakeholders internationally who were knowledgeable about FoP nutrition labelling and Codex, between July 2017 and April 2018. Initial participants were identified through a review of the literature (search terms “codex”, “labelling”, “nutrition”) and public documents, particularly from Codex, WHO and WTO (*n* = 18, across all stakeholder types), and additional participants were identified through snowball sampling (*n* = 10). Potential participants were directly approached via formal email requests to participate, and contact details for participants were obtained through (1) existing contacts of the research team, and (2) publicly available information on institutional websites, including the public record of Codex meetings.

Interviewees included staff from relevant multilateral institutions, including the secretariats of Codex, WHO, WTO, and FAO (*n* = 9); public health and consumer non-government organizations (NGOs) (*n* = 7); public health academics (*n* = 6); and staff from national Codex Contact Points (i.e., the government department responsible for Codex related activities) in countries involved with the Working Group on FoP nutrition labelling (*n* = 6) (Table 1). Twelve respondents were from Low- or Middle-Income Countries (LMICs), the majority from Latin America, and 16 had an explicit public health mandate or professional interest (within multilateral secretariats, academia, and NGOs). Eight invitations to interview were declined or not responded to, including all the invitations sent to industry stakeholders. Nine interviews were conducted in-person, 15 via phone or Skype.

The semi-structured interview schedule was developed based on the theoretical frameworks underpinning the study and piloted before use; interviews were 40 to 60 min in length. The questions focused on: the ongoing Codex discussions regarding FoP nutrition labelling, with respect to: the broader institutional and agenda setting context; framing of the policy issue; policy and governance structures and processes, and the exercise of power by actors within these; and potential content, as well as strengths and limitations, of global guidance from Codex. Detailed notes were written during and immediately after the interview and sent to the interviewee for review and correction. Except for the lack of participation (and thus perspectives) from food industry stakeholders (noted above) we reached a point of data saturation in relation to our core research question regarding influences on decisions about FoP nutrition labelling at the global level.

### 2.3. Data Analysis

The detailed notes from interviews were analyzed using NVivo. We coded for predetermined themes based on our theoretical frameworks, and open-coded in line with our research question. Predetermined themes for coding were: context (historical, nutritional and trade/industry); content recommendations; process; framing of the policy issue; structure and relationships of actors and institutions (including exercise of power); and actor interests. Additional themes arising from the data were: resources; actor influence; implementation; conflict of interest; and agenda setting. We then analyzed the coded data with respect to our analytical frameworks, described above, focusing on the perceived role and impact of Codex guidance and the location of Codex in the global institutional architecture relevant to FoP nutrition labelling, as well as on how influence is exercised at Codex.

This project was approved by the Human Research Ethics Committee of the University of Sydney (Project number 2017/161).

## 3. Results

We found that Codex guidance was perceived as likely to have a high impact on FoP nutrition labelling globally, either positive or negative depending on the nature of the guidance ultimately developed. We also identified a small and highly interconnected “regime complex” surrounding FoP nutrition labelling at the global level and found that influence on Codex discussions is being exerted differentially by actors at the national and global level, particularly by government and industry actors.

### 3.1. Impact of Codex

All the respondents made mention of Codex as a norm-setting institution, with a strong influence on national food regulation globally, as well as on industry standards and regional standards. Countries with limited regulatory capacity (i.e., many LMICs) were identified as likely to adopt Codex guidance without amendment, adding to the impact of Codex decisions on national regulation. Almost all respondents referred to the fact that the (voluntary) standards and guidelines of Codex are referenced by the (binding) Agreements of the WTO, and observed that without a Codex standard, national governments were likely to be vulnerable to challenges at the WTO (more detail in following section).

However, Codex processes were seen by public health actors—particularly academics—to tend towards resulting in guidance that reflected the “lowest common denominator”. Several respondents articulated a contrast with the WHO’s Framework Convention on Tobacco Control, which was seen as a reference for strengthening public health action and leadership; whereas Codex guidance was more commonly seen as a reflection or codification of existing practice. Four public health respondents also identified instances of industry actors (who were often more knowledgeable about detail and implications of Codex guidance than public officials) effectively using Codex strategically: alleging to government actors that certain public health measures were non-compliant with Codex, even when the legal implications of Codex guidance at the national level were unclear.

There was consistent articulation by respondents of the need for Codex guidance to support effective and contextually appropriate approaches to FoP nutrition labelling. All respondents identified the objective of preventing diet-related NCDs as important, and all but a few perceived FoP nutrition labelling as an effective measure to achieve this aim. Almost all respondents indicated Codex guidance done “well” would strengthen and expand FoP nutrition labelling action, because of its role in providing a reference to countries and protection from trade challenges, but there was a preference by many respondents for a “non-prescriptive” guideline. Three respondents from multilateral institutions specifically highlighted the value of broad guidance that identified core principles as a common starting point for developing regulation. However, there was some disagreement about whether it would be appropriate for Codex guidance to stipulate process requirements, with three respondents referring to the EU’s process-oriented guidance on FoP nutrition labelling as a potential model, and others raising concerns that this would place onerous requirements on countries with low resources.

All but a few respondents highlighted the divergence between industry-preferred and public health-preferred approaches to labelling, with public health respondents often characterizing industry-preferred approaches as “complex” or “weak”. All the public health respondents identified a risk that “poor” Codex guidance could limit or constrain policy space for countries desiring to implement innovative, mandatory, and/or strongly interpretive (rather than descriptive) forms of labelling.

### 3.2. The Global Regime Complex for FoP Nutrition Labelling

A regime complex is characterized by partially overlapping and nonhierarchical institutions, including more than one international agreement or authority [27]. The four institutions identified by respondents as core to the global governance of FoP nutrition labelling were Codex, the WTO, the WHO, and the FAO.

Codex was consistently identified as an appropriate forum for international discussions and guidance on FoP nutrition labelling due to its historical responsibilities for standards and guidance on food labelling, and its status as a standards-setting body according to the Agreements of the WTO. However, public health respondents (from NGOs, academia and WHO) raised concerns about the structure of Codex and its appropriateness as a forum for making decisions on NCD prevention policy. Codex was repeatedly described as at the interface between science and politics and/or trade and health. This reflected the tensions between Codex’s status as both an intergovernmental (UN) body, and a science-based standards-setting body, and its dual mandate of promoting trade and protecting consumer health. These were described by many respondents as necessary tensions—reflecting similar tensions faced by national governments in achieving both trade and health objectives—but also as limiting the ability of Codex to fully champion effective public health measures.

Public health respondents indicated that these tensions were compounded by the presence of (food) industry actors as formal participants in Codex decision-making processes, both as observers and as members of country delegations. Almost all public health respondents and those from consumer-oriented NGOs raised concerns regarding conflicts of interest arising from industry participation in Codex decision-making. Almost all public health respondents drew attention to the imbalance of representation between public health actors and food industry actors at Codex, including in the eWG on FoP nutrition labelling, as likely to favor industry preferences in resultant guidance. Several respondents, particularly from multilateral institutions, highlighted that there was also often limited participation by LMIC governments. One respondent suggested that this may result in inappropriate standards for these countries.

There was an interesting juxtaposition, noted above, of Codex as influential but lacking leadership. This was identified as a limitation in responding to the “new” challenge of diet-related NCDs in a forum with an historical responsibility for food safety. Several respondents also pointed out that the nature of risk as historically assessed by Codex processes related to acute and direct risks associated with food safety, and that this was inappropriate in the context of NCDs, where risk is long term and multi-factorial. In line with this, five Codex-affiliated respondents (from national Codex Contact Points and Codex secretariat) indicated that they were doubtful of the effectiveness of FoP nutrition labelling in addressing obesity and NCDs.

All respondents identified the epidemiological and nutrition transition—characterized by an increased prevalence of diet-related NCDs globally—as a key contextual factor leading to a (for some respondents “urgent”) need for nutrition policy action. Two respondents identified the current back of pack nutrient information panels used in most countries, which are already subject to Codex guidance, as inadequate and/or ineffective in promoting healthier diets. One public health respondent specifically drew attention to the difference between a consumer-rights approach to labelling, focused on providing understandable information to consumers (this echoed comments by the 2 consumer NGOs), and a public health approach, which goes beyond this to actively promote healthier choices.

However, almost all respondents intimated that the primary reason that FoP nutrition labelling was on the Codex agenda was concerns about trade and harmonization. These consisted of industry concerns relating to the consistency of requirements in export markets and the concerns raised by government representatives in the WTO, in which the current guidance on supplementary nutrition labelling was implicitly identified as insufficient.

The primacy of concerns about trade and industry related issues was also evident in framing by some respondents (non-public health)—and frames used by industry actors that had been observed by public health respondents—of FoP nutrition labelling as a “restrictive regulatory measure” that was being implemented in unnecessarily diverse approaches that had associated risks of limiting trade. These actors also emphasized the importance of harmonization to reduce barriers to trade. Three respondents (non-public health) emphasized that labelling is only one intervention and by itself would not “solve obesity”.

A significant part of the influence of Codex on national policy was attributed to its reference by the Agreements of the WTO; this relationship was mentioned by all except one respondent. Codex is not the only standards-setting body referenced by the TBT—the Agreement under which specific trade concerns relating to FoP labelling have been raised—but it is the only food standards body that meets requirements. These include being an “open” organization, which involves all relevant stakeholders (in particular, government, industry, and civil society), which was identified by a few non-public health respondents as a specific strength of Codex.

The relationship between the WHO and Codex—and the WHO’s interest in FoP nutrition labelling—was mentioned by all respondents. Eight of the public health respondents mentioned the WHO’s work on nutrient profiling and FoP nutrition labelling as relevant, and indicated that the WHO guideline development related to FoP nutrition labelling should be referenced in the Codex guidance. The WHO was described as a significant point of reference or source of normative guidance on technical issues, with some respondents indicating a role for the WHO in sharing best practice and lessons for health policy development. The WHO was cited by several respondents as having a sole mandate for public health, in contrast to Codex’s dual mandate. However, WHO respondents highlighted that, in contrast to Codex, the WHO does not provide a legal framework. This statement likely refers to the requirement by the Agreements of the WTO that recognized international standards-setting bodies have “open” membership enabling input from all relevant stakeholders (i.e., from all WTO members) [28].

Nine respondents mentioned FAO, and six of these respondents described it as a relative newcomer to issues of nutrition relevant to diet-related NCDs. It was also identified by a few respondents as having more of a trade and industry mandate than a health focus.

All respondents also noted that different Ministries and Departments are sent to represent national governments in these global decision-making bodies, specifically: Agriculture at FAO, Health at the WHO, Trade at the WTO, and a mix at Codex. One respondent highlighted that as a result, countries can end up taking different stances in different forums. For example, at the one might see unanimous support for FoP nutrition labelling from member states and the World Health Assembly, while at the same time, the same countries raise concerns over FoP standards in trade forums. This is exacerbated when national delegations to Codex include industry or food sector representatives. Countries that had existing FoP nutrition labelling schemes in place were identified by several respondents as more likely to be highly engaged in Codex processes.

### 3.3. Power and Influence at Codex

Country delegations were identified consistently as the actual decision makers at Codex (i.e., with voting rights). However, all public health respondents identified industry as strongly influential in Codex decision-making processes. Avenues for influence spanned national and global forums, and sources of influence identified included relational, knowledge-based, and financial resources that enabled participation in decision-making, all of which were seen as imbalanced between public health and industry actors.

#### 3.3.1. Avenues for Influence

Most respondents identified country delegations as most influential at Codex, as they are the actual decision makers. However, respondents also identified avenues for influence by non-government actors, including lobbying and participation in formal decision-making forums, at the national government level as well as directly in Codex (both as part of member state delegations and as official observers).

Many of the LMIC respondents identified strong industry lobbying at the national level as a strategy to influence decisions relating to FoP labelling. Two respondents from Latin American countries described specific instances of “corrupt approaches” to incentivize officials to prevent adoption of policies that would reduce consumption of foods associated with diet-related NCDs, including FoP nutrition labelling and marketing restrictions.

Several public health respondents also observed differences in the targets of industry and public health lobbying at the national level, with industry having access to economic policy actors who were perceived to be more influential than health actors in setting government decision-making. One respondent mentioned that in advocating for FoP nutrition labelling at the regional level she was told to speak to health officials, whereas industry actors lobbying against public health proposals for FoP nutrition labelling spoke to trade and industry officials. One public health respondent had also observed bilateral and “backroom” industry/government negotiations that were not always visible.

This “selection” of targets for lobbying was also identified as one potential driver for locating the mandate for setting FoP standards with Codex, rather than a public health agency such as the WHO. It was noted that national Codex Contact Points and negotiators were often situated in Ministries of Industry/Commerce or Agriculture, and only rarely in Ministries of Health. This was described as often leading to a focus in discussions at Codex on trade and industry aspects of standard and guideline development, rather than health implications. National Standards Committees also often include industry representation. One public health respondent specifically attributed the selection of Codex as the forum for discussion to industry being “afraid” of warning label approaches that could prove to be effective public health measures and therefore likely to reduce profits.

Observers at Codex were identified by 11 of the public health respondents to be heavily weighted towards food industry participation, and one noted that only two of the 15 observers to the initial eWG were non-industry. In addition, the inclusion of industry representatives—and more rarely, non-industry actors—in national Codex delegations was identified by public health respondents as a highly influential avenue, although, the Codex Contact Point representatives interviewed stated that industry representatives did not influence decisions.

Two public health interviewees, however, noted that Codex committees seemed cognizant of the imbalance between industry and public health voices, and had observed committees actively pursuing and attempting to give greater attention to comments by NGO observers at Codex discussions in general.

#### 3.3.2. Sources of Influence

In addition to the relational and access-based sources of influence identified in the previous section, the two main sources of influence described by respondents related to knowledge and financial resources.

Large, high-income countries with their own labelling in place were identified as exercising influence based on their expertise, particularly technical expertise housed in regulatory agencies, and their experience with FoP nutrition labelling development and implementation. However, four public health respondents suggested that Codex member states—particularly LMICs—were not always knowledgeable about the implications of Codex guidance, and industry is often more aware of potential implications and better able to identify and lobby for outcomes that protect their interests. Respondents observed that industry actors were often positioned as technical experts, and used this positioning to engage with decision makers. A few public health respondents noted that in economic and trade-oriented forums (the orientation of many of the relevant Codex decision makers) industry arguments relating to economic impacts were considered more favorably than in public health forums.

In contrast, five public health respondents identified a lack of awareness and knowledge of Codex by the NCD-related public health community as a reason for their limited influence. Unlike industry, relevant public health actors were not familiar with Codex governance structures and avenues for engagement. In fact, two of these respondents suggested that many public health practitioners and researchers were not even aware of discussions in Codex regarding FoP nutrition labelling. Public health actors were thus not well organized to participate and exert influence on the discussions.

Respondents from multilateral organizations emphasized the strong scientific basis that underpinned Codex guidance. Respondents suggested that public health influence was constrained by the limited and only emerging evidence base for the effect of FoP nutrition labelling. However, three public health respondents also identified industry funded research as casting doubt on public health research findings, similar to experiences with tobacco control. While a range of countries have conducted their own research to adopt context specific approaches, it was unclear how Codex processes would take this evidence into account in developing guidance.

Different levels of financial resources available to different actors were also identified by respondents as another source of actor influence on decisions at Codex, but one which was strongly inter-related with relational influence and knowledge.

Three respondents highlighted that development of standards and guidelines require a high level of human resources and technical capacity to develop and implement. As a result, LMICs and small countries can be more influenced by global guidance, because of their low capacity, whereas high-income countries have relatively less need for international guidance, as they have resources and expertise. In the case of FoP nutrition labelling, several respondents pointed out that LMICs are also differentially affected by NCDs, because of insufficient resources within health systems, and due to this may have a greater desire to adopt FoP nutrition labelling schemes as part of NCD prevention strategies.

However, respondents from multilateral institutions and academia pointed out that financial resources also dictate the scale of participation in Codex by member states and observers due to the multiple (in-person) meetings and long time frames for decision-making, which span multiple meetings. LMICs are thus less likely to participate consistently. This leads to an imbalance between influence on, and outcomes of, standards development: countries with fewer resources are more likely to want Codex guidance, but also less likely to have the technical expertise and ability to be present for discussion and development of such guidance.

Respondents from public health also commented on an imbalance between industry and public health participation at Codex. A few respondents noted that industry actors have a significant business motive to be engaged in Codex discussions and devote funding to participate. They also have extensive technical expertise in their particular interest area. In contrast, several respondents noted that public health actors from both government and non-government have a fraction of the resources and are spread very thin, across all Codex issues. There is thus an imbalance of power in both knowledge and in the requirement for engagement. As one public health respondent put it, “industry can pick and choose” the issues they engage on, whereas the Ministry of Health must be across a wide range of food-related public health issues.

## 4. Discussion

Guidance by Codex is likely to be influential in national policy-making due to its established role as an international standards-setting body, which is referenced by the Agreements of the WTO. This study has highlighted that decisions on FoP nutrition labelling at the global level are made in the context of a regime complex and are differentially influenced by national government actors across sectors and income groupings. The industry-oriented framing of issues relating to FoP nutrition labelling at Codex suggests a relatively greater exercise of influence by industry actors through relationships, knowledge, and resources, compared to public health actors. This is facilitated by the governance structure of Codex, which fosters industry influence through multiple although potentially inadvertent mechanisms, such as the cost of participation, and offers multiple national and global avenues for industry influence.

This finding regarding industry influence reflects studies of other issue areas at Codex, such as food safety and food-related definitions [12,14,29,30,31,32]. However, the impact of industry influence may be greater in the NCD-related policy space because of the different interests of industry and public health actors: effective measures have the potential to negatively impact industry profits. Further to the observations of the interviewees in this study, industry actors have strongly opposed national action on FoP nutrition labelling, including through direct lobbying and contesting the evidence in France [33] and Chile [34]. The issue of conflict of interest in nutrition policy-making was highlighted prominently in the WHO’s 2013–2020 Global Action Plan for Prevention and Control of NCDs [35], in which the first nutrition policy option identified is that “member states should consider developing or strengthening national food and nutrition policies…while protecting dietary guidance and food policy from undue influence of commercial and other vested interests”. Conflicts of interest have been identified for many food industry actors regarding policy for addressing diet-related NCDs, as such policies, including FoP nutrition labelling, aims to reduce consumption of unhealthy (often highly profitable) foods. Codex does provide some level of transparency regarding decisions. However, to address the lack of balance in representation and limited public availability of documentation described here, it is likely that the process would be strengthened from a public health perspective with explicit dialogue regarding management of conflicts of interest [36,37,38].

This research also suggests potential for public health actors to increase their influence through strategic participation, coordination, and communication of evidence. Opportunities exist for public health engagement that would be better exploited through coordinated approaches to Codex via national and Observer input. Consideration from a political science perspective suggests that establishment of a focused advocacy coalition [39] including researchers, civil society (including representation of those affected by poor diets), health officials, and donors could foster coordinated public health input into Codex processes regarding FoP nutrition labelling.

Coordinated public health engagement will, however, need to be predicated on improved clarity regarding the objective of FoP nutrition labelling. This was also an implicit issue in the discussions on FoP nutrition labelling in the WTO TBT Committee [8]. The research presented here identified three different perspectives on objectives, ranging from prevention of obesity/NCDs, to promoting healthier choices (more of a social engineering approach) to providing understandable consumer information (more of a liberal individual choice issue). This raises the question of the extent to which the positioning of the objectives of FoP nutrition labelling might influence the likelihood of Codex advice that favors public health objectives. The strong emphasis of Codex on consumer protection means that the “right to know” consumer argument may be more consistent with industry or economic/agriculture Codex Contact Points than the health arguments. In particular, from a public health perspective, FoP nutrition labelling is clearly articulated as a “point of purchase” strategy to encourage changes in food habits beyond informing consumers. Related to this is the challenge of the still-emergent public health evidence as to the best system, which makes it difficult to categorically assess the evidence in relation to (1) whether there are different approaches that might best achieve different objectives (e.g., consumer information compared to behavior change); (2) the risks associated with NCDs; and (3) the impact of FoP nutrition labelling measures on the prevalence of obesity and/or diet-related NCDs. Given this, it seems likely that the best Codex guidance would support contextual adaptation, to foster innovation based on local research.

The location of the discussion of formal global guidance on FoP nutrition labelling within Codex and not the WHO is suggestive of forum shifting. The strong recognition of conflicts of interest regarding nutrition at the WHO mentioned above, as well as the much more curtailed (formal) avenues for industry input suggests that public health interests may be tempered, in favor of industry interests, with the discussion located within Codex. However, this location also recognizes that labelling, as a technical measure, also has economic dimensions which need to be considered by policy makers. Such shifts in food policy decision-making to an economic forum has also been seen with food security policy. Shifting of global policy responsibility to the G20 (similarly, an economic forum) from the Committee on World Food Security (CFS) altered the vision of food security that should ultimately guide global policy-making from an agriculture-oriented definition to an economic one [23].

The regime complexity observed in this study is also reminiscent of that observed in food security, in which the WTO, FAO, CFS, and World Food Program (among others) share responsibility for food security at the global level, but with differing mandates and priorities [40]. One of the risks of this regime complexity is the greater opportunity for powerful actors to create agendas and shift to forums favorable to their interests [21]. On the other hand, however, such complexity can also create new opportunities for less powerful actors. For example, through heightening the role of expert advisers and facilitating negotiation and learning by international civil servants in multilateral institutions [27].

This study has drawn on qualitative policy analysis methods to examine processes and influence in the global governance of FoP nutrition labelling. Limitations of the study include the lack of industry participation in interviews and the limited public availability of meeting documentation. These additional sources of data would have helped to enhance understanding of the intricacies of negotiations within the relevant committees and eWG. In particular, the high proportion of public health oriented respondents has enabled the analysis to focus in detail on the existing consideration of public health nutrition in governance of nutrition labelling, and also to ensure that the analysis is communicated constructively and appropriately to the target audience (one key finding was that there is little awareness of global governance of FoP nutrition labelling among the broader public health nutrition community). However, input from industry would have enabled us to interrogate in more detail the other objectives and agendas that are influencing decisions on FoP nutrition labelling at the global level.

## 5. Conclusions

Guidance from Codex is likely to have a significant impact on global adoption of FoP nutrition labelling. However, current institutional structures within the small and highly interconnected “regime complex” surrounding FoP nutrition labelling may result in a tempering of public health interests, in favor of industry interests. It is likely that the process of development of guidance would be strengthened from a public health perspective with explicit dialogue regarding management of conflict of interest. This research also suggests potential for public health actors to increase their influence through strategic participation in global governance forums, coordination of action and messaging, and targeted communication of evidence. Future policy analysis research is needed to examine the outcomes of these global decision-making processes, and their subsequent implications for national level policy (particularly for countries that already have FoP nutrition labelling initiatives in place).

## Figures and Tables

**Figure 1 nutrients-11-00268-f001:**
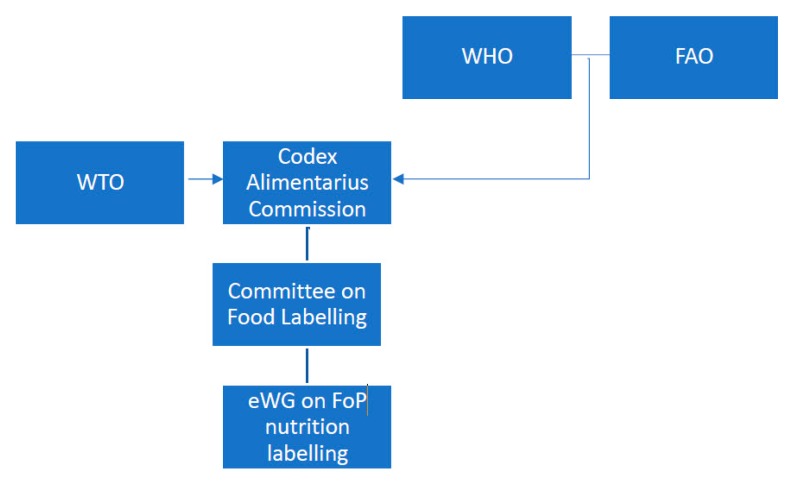
Summary of institutional relationships relevant to FoP nutrition labelling. Abbreviations: WTO–World Trade Organization, WHO–World Health Organization, FAO–Food and Agriculture Organization of the United Nations, eWG–electronic Working Group, FoP–Front of Pack.

**Table 1 nutrients-11-00268-t001:** Summary of interviewees and affiliations.

Country (Income Category) (*n* = 28)	Type of Organization (*n* = 28)	Main Sectoral Interest
High-income countries (*n* = 16)	Multilateral (*n* = 9)	Public health (*n* = 16)
Low- and middle-income countries (*n* = 12, majority from Latin America)	Public health and consumer NGO (*n* = 6)	Economic/industry (*n* = 5)
	Academic (*n* = 6)	Consumer food safety/other nutrition (*n* = 6)
	National governments (*n* = 6)

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
