# Peer review of "Global Governance of Front-of-Pack Nutrition Labelling: A Qualitative Analysis"

_nutrients, 2019, doi:10.3390/nu11020268_

Reviewer 1 Report

The manuscript deals with a relevant and important topic worldwide, related to the implications of international recommendations for the implementation of FOP nutrition labelling policies. The study was well-designed and the . I think the article would be a relevant contribution to the literature from both an academic and political perspective. I include bellow some detailed comments that can contribute to improving the quality of the article. I recommend publication after the comments have been addressed.

Introduction

I think it is important to provide some background information about the different types of FOP nutrition labelling schemes that have been implemented worldwide. In particular, I think it is important to show that different countries have already implemented very different schemes, making harmonization difficult. This can be useful to highlight the differences between industry and government led schemes and the potential tensions with CODEX recommendations.

Line 161. Please provide the guiding questions of the semi-structured interview

Results 

I think it would be good to include example of quotes for the different themes identified in the interviews (a Table can be used for this purpose)

Discussion

I think the discussion could improve by stressing the relevance as FOP nutrition labelling as a public health policy to encourage changes in food habits beyond informing consumers. This is a key element of the opposition between policy makers and industry, particularly at CODEX. In addition, it would be relevant to add some references to news reporting industry opposition to government led FOP nutrition labelling schemes (e.g. the warning labels in Canada, Chile and Uruguay), as it could stress the authors' point regarding the potential influence of the industry in CODEX recommendations.

Finally, it could be good to discuss the potential implications of CODEX recommendations for countries that already have implemented FOP nutrition labelling policies. Can we expect a single FOP nutrition labelling worldwide? Are the recommendations expected to point towards a specific scheme? Should countries modify their policies if they do not comply with the CODEX recommendations? This type of discussion could be useful for readers.

Author Response

REVIEWER 1 COMMENT

 The manuscript deals with a relevant and important topic worldwide, related to the implications of international recommendations for the implementation of FOP nutrition labelling policies. The study was well-designed and the . I think the article would be a relevant contribution to the literature from both an academic and political perspective. I include bellow some detailed comments that can contribute to improving the quality of the article. I recommend publication after the comments have been addressed.

 Introduction

I think it is important to provide some background information about the different types of FOP nutrition labelling schemes that have been implemented worldwide. In particular, I think it is important to show that different countries have already implemented very different schemes, making harmonization difficult. This can be useful to highlight the differences between industry and government led schemes and the potential tensions with CODEX recommendations.

 Response:

We appreciate the reviewer’s suggested and have added to the introduction as follows (line 5-9):

In particular, the schemes that have been implemented vary in terms of 1) designs and content, with some signposting ‘high’ content of nutrients associated with NCD risk and others evaluating and summarising the nutritional quality of products overall; 2) the type of judgement made (positive and/or negative judgements, such as endorsement logos or warning labels); 3) implementation mode (voluntary or mandatory) [1]. This lack of harmonization has resulted in the need for food industry actors to cater to different labelling requirements in different markets…”

 Line 161. Please provide the guiding questions of the semi-structured interview

 Response:

The questions of the semi-structured interview are summarised in lines 181-184:

“The questions focussed on: the ongoing Codex discussions regarding FoP nutrition labelling, with respect to: the broader institutional and agenda setting context; framing of the policy issue; policy and governance structures and processes, and the exercise of power by actors within these; and potential content, as well as strengths and limitations, of global guidance from Codex.”

Results 

I think it would be good to include example of quotes for the different themes identified in the interviews (a Table can be used for this purpose)

Response:

We appreciate the reviewer’s suggestion. However, we are unable to present direct quotes because of the nature of the data collection (as stated, we took detailed notes, which were then approved by interviewees as an accurate report of the discussion, but they are not in participants words and so not appropriate for quoting).

Discussion

I think the discussion could improve by stressing the relevance as FOP nutrition labelling as a public health policy to encourage changes in food habits beyond informing consumers. This is a key element of the opposition between policy makers and industry, particularly at CODEX.

Response:

We appreciate the reviewer’s suggestion and have now added this point specifically to our discussion of the need for improved clarity of the objective of FoP nutrition labelling, as follows (line 478-480):

In particular, from a public health perspective, FoP nutrition labelling is clearly articulated as a ‘point of purchase’ strategy to encourage changes in food habits beyond informing consumers.”

In addition, it would be relevant to add some references to news reporting industry opposition to government led FOP nutrition labelling schemes (e.g. the warning labels in Canada, Chile and Uruguay), as it could stress the authors' point regarding the potential influence of the industry in CODEX recommendations.

Response:

We appreciate the reviewer’s suggestion and have now added commentary on this to our discussion of conflicts of interest, lines 445-464, as follows:

Further to the observations of the interviewees in this study, industry actors have strongly opposed national action on FoP nutrition labelling, including through direct lobbying and contesting the evidence in France [2] and Chile [3].

Finally, it could be good to discuss the potential implications of CODEX recommendations for countries that already have implemented FOP nutrition labelling policies. Can we expect a single FOP nutrition labelling worldwide? Are the recommendations expected to point towards a specific scheme? Should countries modify their policies if they do not comply with the CODEX recommendations? This type of discussion could be useful for readers.

Response:

We appreciate the reviewer’s suggestion, and have partially addressed this in our comments in the discussion (lines 486-489) regarding the need for Codex guidance to support contextual adaptation. But a detailed discussion of the implications of potential Codex guidance is outside of the scope of our research. We have however added new text on the need for future research to address this important issue (lines 529-532), as follows:

Future policy analysis research is needed to examine the outcomes of these global decision making processes, and their subsequent implications for national level policy (particularly for countries that already have FoP nutrition labelling initiatives in place).

Reviewer 2 Report

Review Thow et al 2018

Interesting paper! To rate its value, it is important to have more insight in a detailed description the group of 28 stakeholders that were interviewed. Industry representatives are missing, yet results are affecting them. How would this be different, if they were part of the study population.

Abstract

The term “complex regime “ is used. But what is exactly meant by this? A description would help.

The introduction is informative, interesting to read.

L61: what is meant by ‘alternatives’ in this sentence, alternative for what?

Materials and Methods

28 stakeholders were interviewed. Some details of the study group are missing.

L146 participants were identified through review of literature and public documents: what where the search terms?

Unfortunately none of them were from industry (L160). How many stakeholders from industry were invited to be interviewed? This not clear form the text.

A more detailed description of the stakeholders is missing. How are the different organizations represented, pictured in Fig 1 and described (L57-71). Table of a figure could give more insight into this.

Have you reached data-saturation?

Results

In the results it is not always clear where the respondents are from (see remark at Methods section). Although many seem to come from ‘public health’.

L214 respondents from ‘multilateral institutions’. Where are these respondents from, how do they fit in the picture (related to remark above)?

L348 …only two of the Observers to eWG were non-industry: How many observers where there in total?

Discussion

A Critical evaluation of the effect of the composition of the study group:

 It seemed that most of the stakeholders are public health related. (L152-160) How does that affect the results?

L482 mentioned lack of industry participation as study limitation: how did this affect the results? Can you elaborate?

L416 … at the global are made: add ‘level’ after global

L420 … This is facilitated the governance: omit ‘is’

Author Response

REVIEWER 2 COMMENTS

 Interesting paper! To rate its value, it is important to have more insight in a detailed description the group of 28 stakeholders that were interviewed. Industry representatives are missing, yet results are affecting them. How would this be different, if they were part of the study population.

 Response:

We appreciate the reviewer’s positive comments. As detailed below, we have added more detailed description of the stakeholders interviewed, and strengthened the commentary on the lack of industry participation.

Abstract

The term “complex regime “ is used. But what is exactly meant by this? A description would help.

Response:

We appreciate the reviewer’s comment and have now clarified in the abstract as follows:

“, a small and highly interconnected ‘regime complex’ of international institutions surrounds FoP nutrition labelling at the global level…”

The term is defined in full in the body of the paper.

The introduction is informative, interesting to read.

L61: what is meant by ‘alternatives’ in this sentence, alternative for what?

Response:

We have now clarified in text as follows:

“each Member ‘may be accompanied by one or more alternatives [i.e. additional people who can act as representative]…”

Materials and Methods

28 stakeholders were interviewed. Some details of the study group are missing.

L146 participants were identified through review of literature and public documents: what where the search terms?

Response:

We appreciate the reviewers request for clarification. We have amended the sentence as follows:

“Initial participants were identified through a review of the literature (search terms ‘codex’, ‘labelling’, ‘nutrition’) and public documents, particularly from Codex, WHO and WTO…”

Unfortunately none of them were from industry (L160). How many stakeholders from industry were invited to be interviewed? This not clear form the text. A more detailed description of the stakeholders is missing. How are the different organizations represented, pictured in Fig 1 and described (L57-71). Table of a figure could give more insight into this.

Response:

We appreciate the reviewer’s suggestion, and have developed a summary table to complement the text in lines 155-163 on the detail of stakeholder composition

Have you reached data-saturation?

Response:

We appreciate the reviewer’s question – we did reach a point of data saturation, and have now indicated this in the methods (line 188-188):

Except for the lack of participation (and thus perspectives) from food industry stakeholders (noted above) we reached a point of data saturation in relation to our core research question regarding influences on decisions about front of pack nutrition labelling at the global level.

Results

In the results it is not always clear where the respondents are from (see remark at Methods section). Although many seem to come from ‘public health’.

L214 respondents from ‘multilateral institutions’. Where are these respondents from, how do they fit in the picture (related to remark above)?

Response:

We appreciate the reviewer’s concern. The multilateral institutions represented are detailed in the methods (lines 162-163). We have reported the findings by interviewee type throughout the results, where these differentiations were meaningful.

L348 …only two of the Observers to eWG were non-industry: How many observers where there in total?

Response:

We appreciate the reviewer picking up on this point. We have now clarified as follows:

“…one noted that only two of the 15 Observers to the initial eWG were non-industry.”

Discussion

A Critical evaluation of the effect of the composition of the study group:

·        It seemed that most of the stakeholders are public health related. (L152-160) How does that affect the results?

·        L482 mentioned lack of industry participation as study limitation: how did this affect the results? Can you elaborate?

 Response:

We appreciate the reviewer’s suggestion and have added more reflection on the impact of the composition of the study group on the findings, lines 514-20, as follows:

In particular, the high proportion of public health oriented respondents has enabled the analysis to focus in detail on the existing consideration of public health nutrition in governance of nutrition labelling, and also to ensure that the analysis is communicated constructively and appropriately to the target audience (one key finding was that there is little awareness of global governance of FoP nutrition labelling among the broader public health nutrition community). However, input from industry would have enabled us to interrogate in more detail the other objectives and agendas that are influencing decisions on FoP nutrition labelling at the global level.

L416 … at the global are made: add ‘level’ after global

Response:

Amended

L420 … This is facilitated the governance: omit ‘is’

Response:

Amended (“This is facilitated by…”)